# Significant Variations of Thermospheric Nitric Oxide Cooling during the Minor Geomagnetic Storm on 6 May 2015

Zheng Li [1,2,*], Meng Sun [1], Jingyuan Li [1], Kedeng Zhang [3], Hua Zhang [1], Xiaojun Xu [2] and Xinhua Zhao [4]

1   Institute of Space Weather, Nanjing University of Information Science and Technology, Nanjing 210044, China; 20191216003@nuist.edu.cn (M.S.); jingyuanli@nuist.edu.cn (J.L.); 002461@nuist.edu.cn (H.Z.)
2   State Key Laboratory of Lunar and Planetary Science, Macau University of Science and Technology, Macau 999078, China; xjxu.ncu@gmail.com
3   Department of Space Physics, School of Electronic Information, Wuhan University, Wuhan 430072, China; ninghe_zkd@whu.edu.cn
4   State Key Laboratory of Space Weather, National Space Science Center, Chinese Academy of Sciences, Beijing 100190, China; xhzhao@spaceweather.ac.cn
*   Correspondence: zli@nuist.edu.cn

**Abstract:** Using observations by the SABER (Sounding of the Atmosphere using Broadband Emission Radiometry) instrument on board the TIMED (Thermosphere-Ionosphere-Mesosphere Energetics and Dynamics) satellite and simulations by the TIEGCM (Thermosphere-Ionosphere-Electrodynamics General Circulation Model), we investigate the daytime variations of thermospheric nitric oxide (NO) cooling during the geomagnetic storm on 6 May 2015. The geomagnetic storm was minor, as the minimum Dst was −28 nT, the maximum Kp was 5+ and the maximum AE was 1259 nT. However, significant enhancements of peak NO cooling rate and prominent decreases in the peak NO cooling altitude were observed from high latitudes to low latitudes in both hemispheres on the dayside by the SABER instrument. The model simulations underestimate the response of peak NO cooling and have no significant variation of the altitude of peak NO cooling rate on the dayside during this minor geomagnetic storm. By investigating the temporal and latitudinal variations of vertical NO cooling profiles inferred from SABER data, we suggest that the horizontal equatorward winds caused by the minor geomagnetic storm were unexpectedly strong and thus play an important role in inducing these significant daytime NO cooling variations.

**Keywords:** nitric oxide cooling; thermosphere; geomagnetic storm





## 1. Introduction

Geomagnetic storms usually refer to a large amount of energy from the solar wind deposited in the magnetosphere-ionosphere-thermosphere system mainly via particle precipitation and Joule heating. A geomagnetic storm can last from hours to several days and cause significant disturbances in neutral temperature, composition, density, as well as ionospheric total electron content and critical frequency. The large amount of energy injected into the polar thermosphere/ionosphere enhances the equatorward winds, and thus makes the significant disturbances in thermosphere/ionosphere a global phenomenon [1–10].

The thermospheric temperature increases due to the deposition of energy during a geomagnetic storm. The cooling mechanisms in the thermosphere include heat conduction and infrared radiation from $CO_2$, NO, and O [11]. NO is a minor constituent in the mesosphere and lower thermosphere, and is created through the reaction (R1) between excited-state atomic nitrogen $N(^2D)$ and molecular oxygen and the reaction (R2) between ground-state atomic nitrogen $N(^4S)$ and molecular oxygen [12,13]. However, the NO 5.3 µm emission, which was created by the collision between NO and energetic atomic oxygen (R3), is highly correlated with the global Joule heating power with a time lag of 10 h and

appears to be the dominant thermospheric cooling agent in regulating the thermospheric temperature during geomagnetic storms [14–17]. Therefore, NO emission is considered a "natural thermostat" that contributes to the thermosphere recovery from the effects of a geomagnetic storm [14,18].

$$N(^2D) + O_2 \rightarrow NO + O \tag{1}$$

$$N(^4S) + O_2 \rightarrow NO + O \tag{2}$$

$$NO + O \rightarrow NO + O + h\nu_{5.3\mu m} \tag{3}$$

A number of studies have been carried out on the effects and behaviors of NO cooling under the conditions of intense geomagnetic storms using observations and simulations over the past several decades. Siskind et al. [19] attributed the NO density increase at midlatitudes to the local increased temperatures at the storm times, rather than horizontal transport of NO from high latitudes. Using the SABER data, Mlynczak et al. [14,18] found that the NO zonal mean radiance increased by a factor of 5 to 7 and the energy radiated by NO accounted for 50% of the total input energy to the upper atmosphere during the geomagnetic storms of April 2002. Lu et al. [17] found that the NO infrared emission accounts for the dissipation of nearly 80% of the Joule heating energy input during the six selected intense geomagnetic storms by utilizing SABER measurements and TIEGCM simulations. Using CHAMP, GRACE and SABER data, Lei et al. [20] found that there was an "overcooling" effect in thermospheric neutral densities caused by elevated NO cooling rate during the recovery phase of the 2003 October storms. Compared with SABER measurements, Sheng et al. [21] found that the TIEGCM overestimates the NO cooling power at low latitudes and underestimates it at high latitudes during the geomagnetic storm of 5 April 2010. Chen and Lei [22] carried out a series of controlled numerical experiments of TIEGCM to explore the processes responsible for the neutral density overcooling during the recovery phases of the October 2003 storms. For the superstorm event of 7–12 November 2004, Bharti et al. [23] found that the SABER-retrieved NO cooling rate at a local site suggests an enhancement with the peak emission rate closely correlated to the progression of the storm, and the peak emission altitude of the NO cooling rate moves upward during the main phase of the storm. Utilizing the SABER measurements and TIEGCM outputs, Li et al. [24] investigated the behaviors of NO cooling during the 15 May 2005 intense geomagnetic storm and suggested that the horizontal equatorward transport plays an important role in inducing the upward shift of the NO cooling peak altitude on the nightside and the downward shift of it on the dayside. Bag [25] studied the diurnal variation of height-distributed nitric oxide radiative emission during the November 2004 superstorm by using SABER data. In addition, they also studied storm-time hemispheric asymmetry in NO radiative cooling during intense geomagnetic storms and suggested that the storm-time meridional wind could play an important role resulting in the hemispheric asymmetry [26].

This study focuses on the behaviors of NO cooling during a minor geomagnetic storm. On 6 May 2015, there was a fairly weak geomagnetic storm, as the minimum Dst was only −28 nT. Unexpectedly, significant enhancements of peak NO cooling rate and prominent decreases in the peak NO cooling altitude were observed from high latitudes to low latitudes in both hemispheres on the dayside by the SABER instrument. The key mechanism of the variability of NO cooling is discussed in this paper by analyzing variations of vertical profiles of NO cooling rate.

## 2. Data and Model

### 2.1. TIMED/SABER Observations

The TIMED satellite was launched on 7 December 2001 into a near-circular orbit at 625 km with an inclination of 74°. The SABER instrument on board the TIMED is a 10-channel infrared radiometer that scans the Earth's limb from the 400 km tangent height down to about 20 km with a vertical resolution of 2 km. The spectral coverage of the instrument is from 1.27 to 15.4 μm, including NO infrared emission at 5.3 μm [14,27]. SABER covers the Earth's geographic latitudes asymmetrically during any 60 days due to

its anti-Sun view and so the latitudinal coverage of the SABER ranges from about 52° in one hemisphere to 83° in the other [18,28]. During the geomagnetic storm of 6 May 2015, SABER was in a south viewing mode and covering a latitude range between 52° N and 83° S. NO cooling rates derived from the SABER version 2.0 data set are utilized in this study, and the features of SABER NO cooling flux are studied by Flynn et al. [29] using the EOF method.

### 2.2. TIEGCM Simulations

The National Center for Atmospheric Research (NCAR) TIEGCM is a first-principles, time-dependent, three-dimensional model that solves the fully coupled, nonlinear, hydro-dynamic, thermodynamic, and continuity equations of the neutral gas self-consistently with the ion energy, ion momentum, and ion continuity equations as well as the neutral wind dynamo [30–32]. The horizontal resolution of the TIEGCM v2.0 is 2.5° in geographic latitude by 2.5° in geographic longitude and the vertical grid is pressure coordinates extending from ~97 to ~500 km, with a vertical resolution of a quarter scale height. The TIEGCM is driven by solar extreme ultraviolet and ultraviolet spectral fluxes parameterized by the $F_{10.7}$ index [33], geomagnetic forcing represented by high-latitude precipitation and convection pattern [34,35] driven by the Kp index, and diurnal and semidiurnal tidal inputs at the low boundary based on the global scale wave model (GSWM) [36]. The NO 5.3 μm emission is calculated in the TIEGCM based on the formulation given by Kockarts [37], and the features of model NO cooling flux are studied by Li et al. [38] using the EOF method. The TIEGCM data used to compare with the SABER measurements in this study are sampled at the SABER measurement locations.

### 3. Results

### 3.1. Solar Radiation and Interplanetary and Geomagnetic Conditions

There was a minor geomagnetic storm (Kp = 5) on 6 May 2015. Figure 1 shows several parameters and indices that describe the solar radiation and interplanetary and geomagnetic conditions for 5–7 May 2015: (a) the $F_{10.7}$ index, (b) solar wind speed, (c) the interplanetary magnetic field (IMF) $B_z$ and $B_y$ components, (d) Dst index, (e) Kp index, and (f) the AE index. The jump in solar wind speed and large changes in the IMF components at ~02 UT indicate the arrival of a solar wind shock. The sheath ahead of the driving interplanetary coronal mass ejection (ICME) passed Earth for the next four hours. The magnetic cloud structure with a dominant and slowly rotating IMF $B_y$ component affected geospace from 12 UT on 6 May through 21 UT on 7 May 2015. See magnetic cloud listing at http://www.srl.caltech.edu/ACE/ASC/DATA/level3/icmetable2.htm (accessed on 1 March 2022) and explanations in Richardson and Cane [39].

Solar EUV, as indicated by the $F_{10.7}$ index, increased continuously from ~128 to ~147 sfu for the period of 5–7 May 2015. As the solar wind shocks arrived at Earth around 0200 UT on 6 May, solar wind speed showed sharp increases and then reached its peaks of 480 km/s at 0625 UT. Associated with the solar wind shocks arriving, the IMF Bz sharply reached near −12 nT southward and turned northward after 3 h. After oscillating around 0 nT between ~0700 and ~1300 UT on 6 May, the $B_z$ turned southward sharply, reached its minimum of −13 nT, and kept southward for about 5 h.

The Dst index, which is widely used to characterize the level and phase of geomagnetic activities, increased on early 6 May, and decreased continuously in general and reached its minimum of −28 nT at 1900 UT. According to the classification by Gonzalez et al. [40], the geomagnetic activities with a Dst index larger than −30 nT would not be classified as even a weak geomagnetic storm. However, both Kp and AE showed prominent enhancements after the magnetic cloud arrival, and Kp and AE reached their maximum of 5+ and 1259 nT, respectively, around 1400 UT on 6 May. An event with Kp = 5 is rated as a minor geomagnetic storm on the NOAA storm scale.

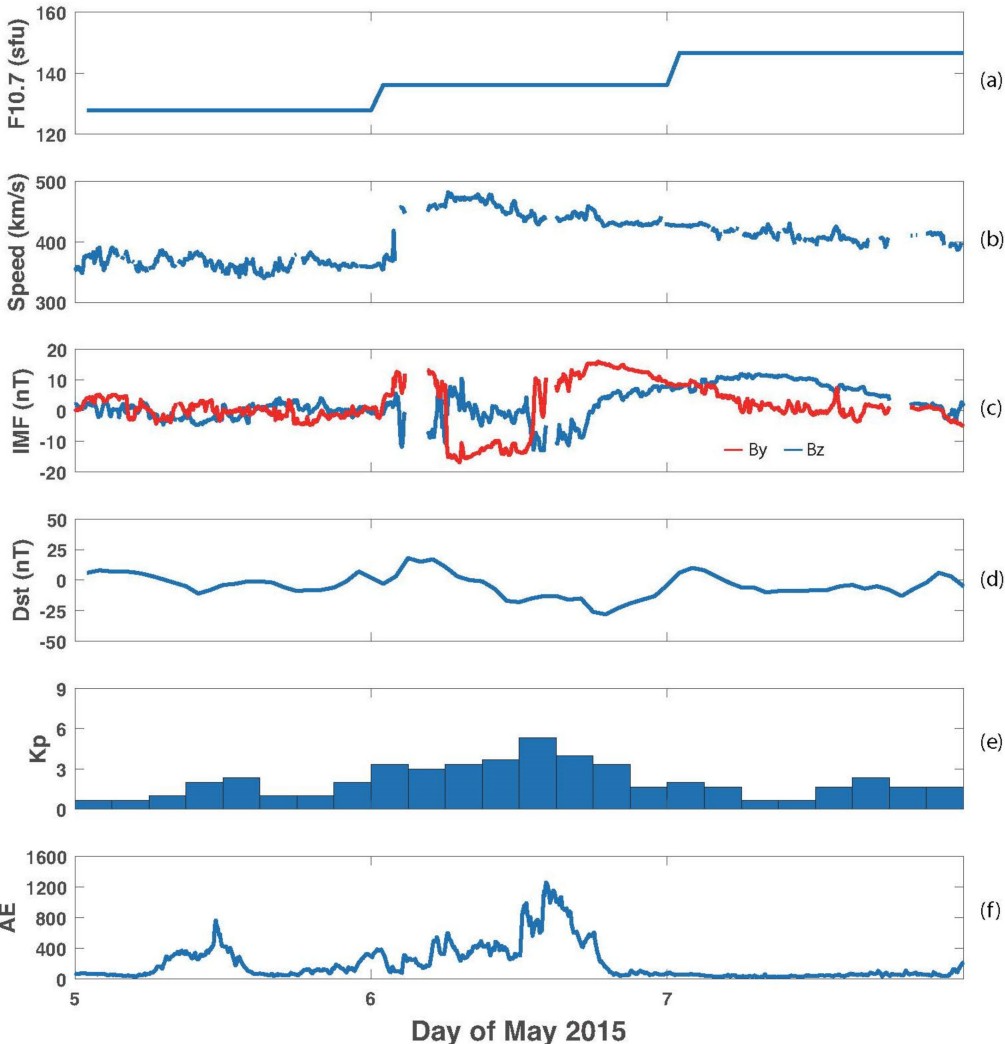

**Figure 1.** Solar radiation and interplanetary and geomagnetic conditions for the period of 5–7 May 2015. (**a**) $F_{10.7}$ index. (**b**) Solar wind speed. (**c**) IMF $B_z$ (blue) and $B_y$ (red). (**d**) Dst index. (**e**) Kp index. (**f**) AE index. IMF = interplanetary magnetic field.

### *3.2. NO Cooling Variations on the Dayside*

#### 3.2.1. Peak NO Cooling Rate

Figure 2 shows the peak NO cooling rate ($W/m^3$) on the dayside (~1330 LT) derived from (a) SABER measurements and (b) TIEGCM simulations for the period of 5–7 May 2015. The SABER viewing geometry favors the Southern Hemisphere during this interval, so it is not surprising to see an NO emission shock response in the Southern Hemisphere just after shock arrival (Figure 2a). Just after mid-day, there is an emission enhancement in the northern hemisphere extending to low latitudes. By the end of the day, we can see that the SABER-observed peak NO cooling rate showed significant enhancements in both hemispheres responding to this weak storm. Between late 6 May and early 7 May, the SABER-observed peak NO cooling rate increased significantly from high latitudes to low latitudes, and it maximized at ~$5.3 \times 10^{-8}$ $W/m^3$ around 38° N at the end of 6 May. Model calculations from the TIEGCM in Figure 2b show a similar trend of peak NO cooling rate with SABER observations to the geomagnetic storm, generally. However, the TIEGCM greatly underestimates the peak NO cooling rate during the weak geomagnetic storm, with a maximum model peak NO cooling rate of only $2.5 \times 10^{-8}$ $W/m^3$.

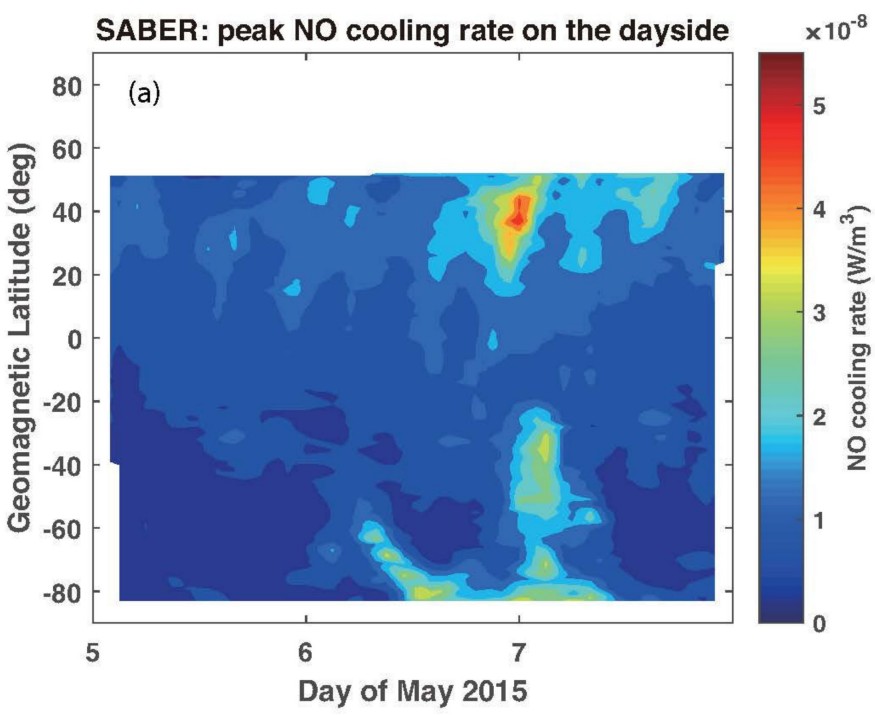

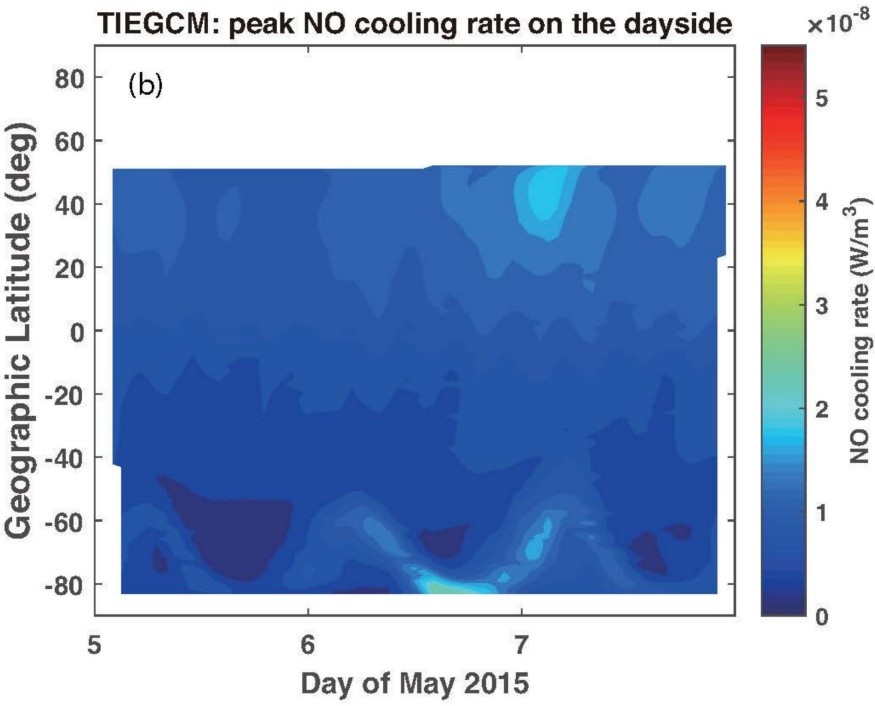

**Figure 2.** Keogram of peak NO cooling rate (W/m³) obtained from (**a**) SABER measurements and (**b**) TIEGCM simulations at about 13:30 LT for the period of 5–7 May 2015.

### 3.2.2. Peak NO Cooling Altitude

The SABER-observed NO cooling rate and TIEGCM simulated the altitude of peak NO cooling rate on the dayside (~1330 LT) during 5–7 May 2015, as presented in Figure 3a,b. From Figure 3a, we can see that the altitude of peak NO cooling rate stayed around 145 km at local noon at low and middle latitudes in the Southern Hemisphere during geomagnetic quiet conditions. Associated with the enhancements of the SABER-observed NO cooling rate between late 6 May and early 7 May, the significant decreases in the altitude of peak NO

cooling rate can be observed from high latitudes to low latitudes. The maximum decrease in peak NO cooling rate can reach 40 km at low latitudes. As Figure 3b shows, there is no significant difference in the altitude of NO cooling rate between before and after the weak geomagnetic storm, which means the TIEGCM failed to capture the altitude variations of peak NO cooling rate during this weak geomagnetic storm. In addition, comparing Figure 3a,b the significant differences for the altitude of the maximum NO cooling rate between the SABER measurements and TIEGCM simulations can be observed at local noon at low and mid latitudes during geomagnetic quiet conditions, which agree with the statistical results by Li et al. [41].

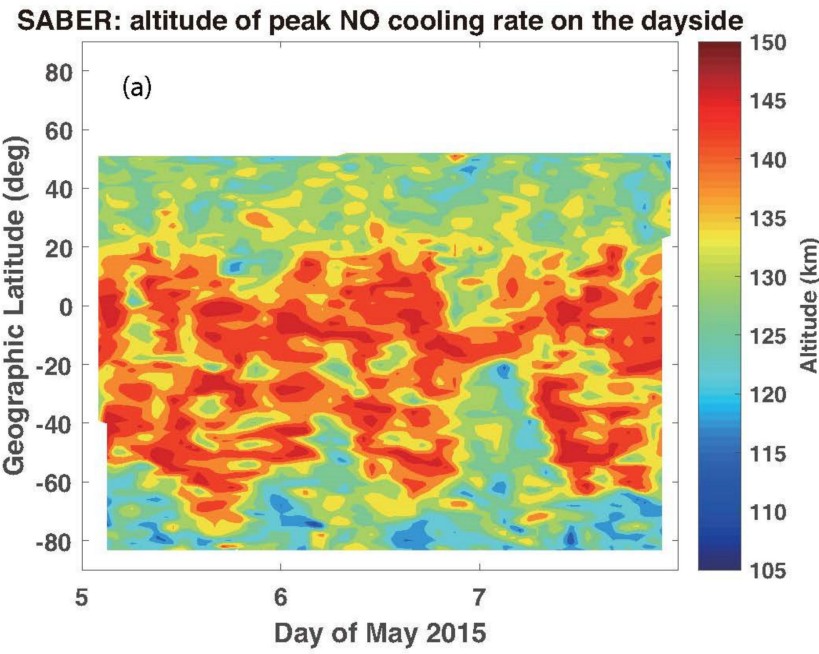

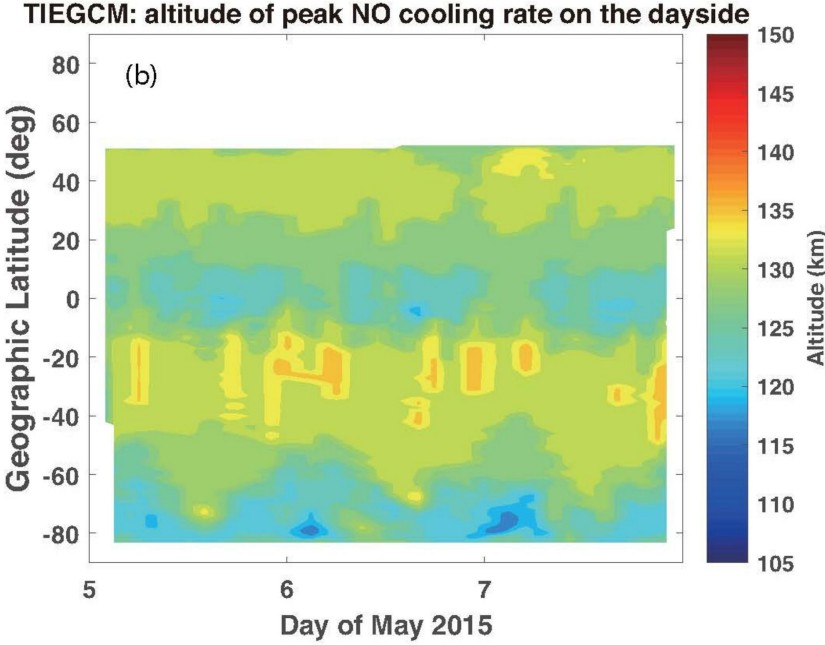

**Figure 3.** Keogram of altitude of peak NO cooling rate obtained from (**a**) SABER measurements and (**b**) TIEGCM simulations at about 13:30 LT for the period of 5–7 May 2015.

## 4. Discussion

The geomagnetic storm that occurred on 6 May 2015 was fairly weak, as the minimum Dst, the maximum Kp, and the maximum AE were −28 nT, 5+, and 1259 nT, respectively. However, both the peak NO cooling rate and its altitude inferred from the SABER data showed unexpectedly large variations responding to this minor storm. On the other hand, the TIEGCM does not respond adequately. On the variation of peak NO cooling altitude during magnetic storms, Li et al. [24] found that there were significant increases in the SABER-derived NO cooling peak altitude at low latitudes on the nightside during the storm main phase and significant decreases in the altitude at low latitudes on the dayside during the storm recovery of the 15 May 2005 intense geomagnetic storm. They also found that the increases and decrease in the NO cooling peak altitude at low latitudes during the intense geomagnetic storm were caused by an unusual double-peak structure in the vertical NO cooling profiles and suggested that the storm-enhanced horizontal equatorward transport played an important role in inducing these significant variations of the NO cooling peak altitude. Late on 6 May 2015, prominent decreases in the altitude of peak NO cooling rate derived by SABER measurements are observed from high latitudes to low latitudes in both hemispheres. In this section, we will discuss the details and mechanisms of these prominent decreases in peak NO cooling rate by analyzing the temporal variations and the latitudinal variations of NO cooling vertical profiles.

Figure 4a–f provide the vertical profiles of NO cooling rate (W/m$^3$) around ~20° S latitude obtained from the SABER data (blue lines) and TIEGCM outputs (red lines) on the dayside from 20:53 UT on 6 May to 06:33 UT on 7 May 2015. At 20:53 UT on 6 May 2015, the low-latitude vertical profiles of NO cooling rate shown in Figure 4a were under weakly disturbed conditions, although Figures 2 and 3 suggest there are larger disturbances at higher latitudes. The SABER-derived peak NO cooling altitude was near 145 km, and the model overestimates the NO cooling rate between 115 and 130 km and underestimates it above 130 km. At 22:30 UT as represented in Figure 4b, the noticeable variation is an enhancement of NO cooling rate between 115 and 130 km observed by the SABER instrument. The enhancement reached ~$1.6 \times 10^{-8}$ W/m$^3$ at around 120 km and thus "merged the normal peak" (that means the new peak of NO cooling rate induced by the enhancement replaced the peak of NO cooling rate, which was at a level under the quiet time condition) at 01:42 UT on 7 May, as observed by the SABER measurement shown in Figure 4c. From Figure 4d, we can see that the enhancement of SABER-derived NO cooling rate became weaker on 03:18 UT, and a double-peak structure of the NO cooling rate occurred with one peak at ~150 km and another at ~110 km, respectively. In the following 3 h (Figure 4e,f), the NO cooling enhancement between 105 and 130 km decreased and returned to the quiet time level. On the other hand, the TIEGCM-simulated NO cooling rate did not show the process and the double-peak structure at the lower thermosphere.

Figure 5a–h show the vertical profiles of the NO cooling rate (W/m$^3$) obtained from the SABER data and TIEGCM outputs from ~77° S to ~20° S at about 03:00 UT on May 5 (during quiet time) and May 7 (during storm). From Figure 5a–f, we can see that the significant enhancements of NO cooling rate were observed from high latitudes to low latitudes by the SABER instrument, while the TIEGCM-simulated NO cooling increased prominently at high and mid latitudes and slightly at low latitudes. Compared with peak NO cooling altitude during geomagnetic quiet time, the enhancement of SABER-derived NO cooling during the minor geomagnetic storm greatly changed the peak NO cooling altitude, especially at mid and low latitudes, where the peak NO cooling altitude during geomagnetic quiet conditions stayed at ~145 km. It is noticeable that the SABER-derived peak NO cooling altitude decreased from ~130 km at high and mid latitudes to ~110 km at low latitudes during the storm. These results indicate that the storm-induced horizontal transport of NO density from high latitudes to lower latitudes became relatively weak at low latitudes and moved downward to a lower altitude and thus caused the peak altitude decrease in NO cooling rate.

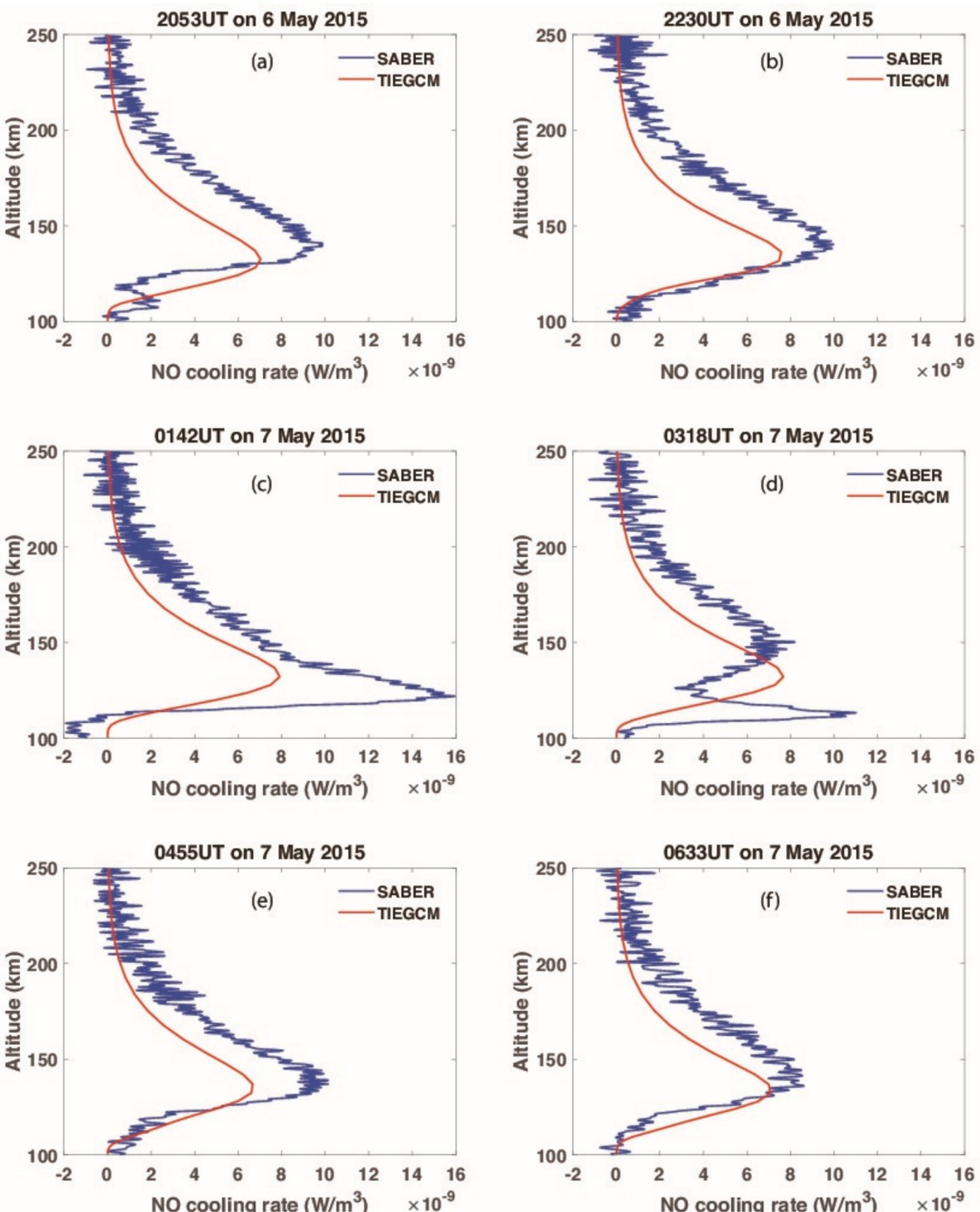

**Figure 4.** Vertical profiles of the NO cooling rate (W/m$^3$) around ~20° S latitude obtained from the SABER data (blue lines) and TIEGCM outputs (red lines) on the dayside from 20:53 UT on 6 May to 06:33 UT on 7 May 2015. The specific time can be seen in the title of each subfigures (**a**–**f**).

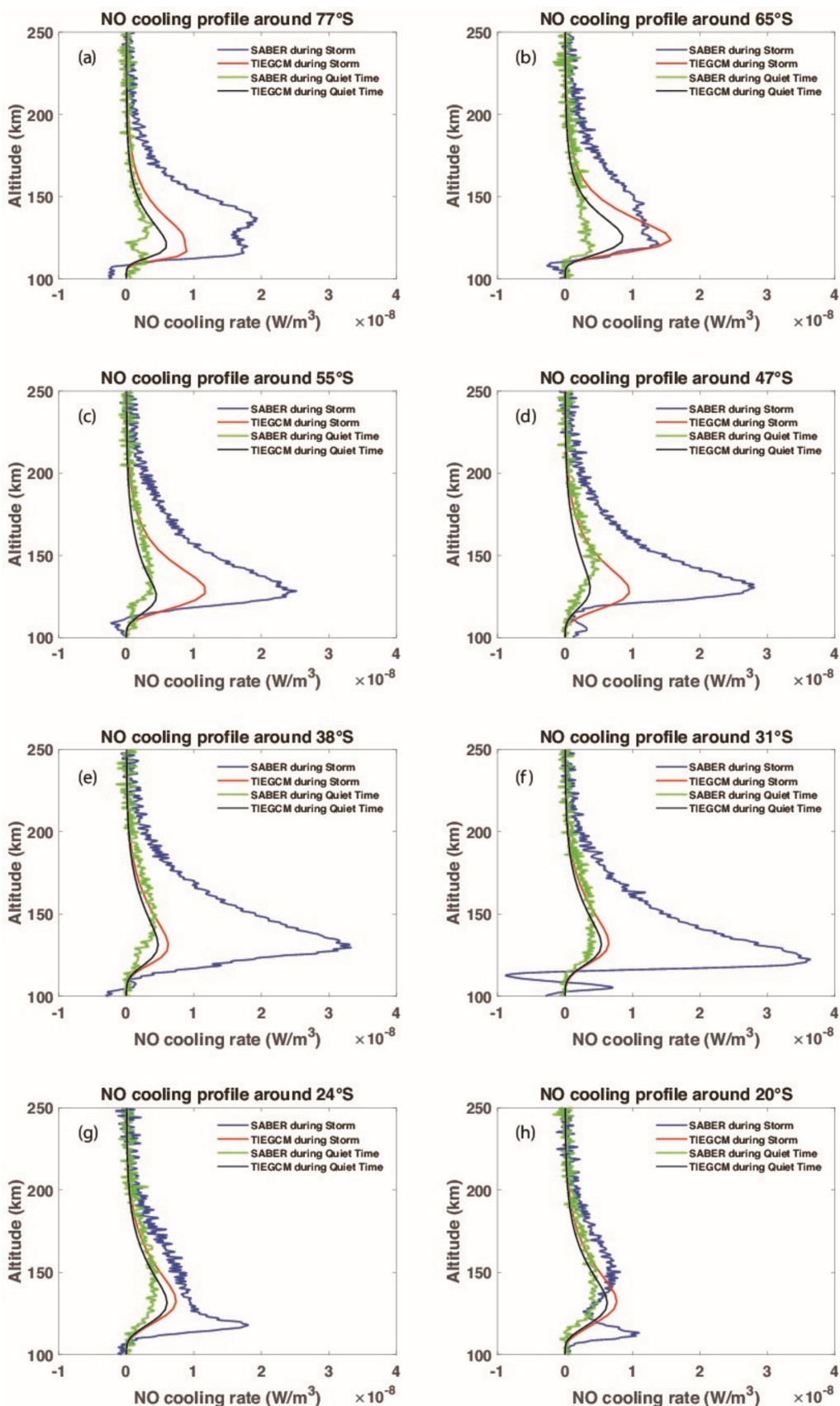

**Figure 5.** Vertical profiles of the NO cooling rate (W/m³) obtained from the SABER data and TIEGCM outputs from ~77° S to ~20° S at about 03:00 UT on 5 May (during quiet time) and 7 May (during storm). The specific latitude can be seen in the title of each subfigures (**a–h**).

Li et al. [24] reported the significant increases in the SABER-derived NO cooling peak altitude on the nightside and significant decreases in the altitude on the dayside during the 15 May 2005 intense geomagnetic storm. In their study, the significant variations of SABER-derived NO cooling peak altitude mainly occurred at low latitudes, and the storm-enhanced horizontal equatorward transport was suggested to be a dominating factor. During the minor geomagnetic storm of 6 May 2015, a significant decrease in peak NO cooling altitude can be observed from high latitudes to low latitudes on the dayside by SABER measurements. By investigating the temporal and latitudinal variations of vertical NO cooling profiles inferred from SABER measurements, we suggest that the horizontal equatorward transport of NO caused by the minor geomagnetic storm was unexpectedly strong and thus significantly changed the daytime peak NO cooling rate and its altitude.

## 5. Conclusions

In this paper, daytime variations of thermospheric NO cooling during the 6 May 2015 geomagnetic storm have been investigated by utilizing TIMED/SABER measurements and NCAR's TIEGCM simulations. The geomagnetic storm (Kp = 5) is rated as a minor geomagnetic storm on the NOAA storm scale. However, a significant enhancement of peak NO cooling rate and a prominent decrease in peak NO cooling altitude can be observed from high latitudes to low latitudes in both hemispheres by the SABER instrument. On the other hand, model simulations of the peak NO cooling rate and its altitude do not show enough response to this minor geomagnetic storm.

An interesting question is what caused these unexpected variations of thermospheric NO cooling during the 6 May 2015 geomagnetic storm. By investigating the temporal and latitudinal variations of vertical NO cooling profiles inferred from SABER data, we suggest that the storm-induced equatorward winds were so strong that not only can they reach the low latitudes but they also last until the day after the storm. These equatorward winds transported the NO and heat from high latitudes to low latitudes at the lower thermosphere, and thus significantly changed the daytime peak NO cooling rate and its altitude. On the other hand, these behaviors are not captured well in the TIEGCM. Based on the results of this paper, further investigations for improving the model capability are needed in the future.

**Author Contributions:** Conceptualization, Z.L., M.S. and J.L.; methodology, Z.L.; software, Z.L.; validation, M.S., J.L. and K.Z.; formal analysis, Z.L.; investigation, Z.L.; resources, Z.L.; data curation, Z.L., H.Z. and X.Z.; writing—original draft preparation, Z.L.; writing—review and editing, H.Z., X.X. and X.Z.; visualization, Z.L.; supervision, Z.L.; project administration, Z.L., J.L. and K.Z.; funding acquisition, Z.L. All authors have read and agreed to the published version of the manuscript.

**Funding:** This research was funded by the National Natural Science Foundation of China (grant number 42074183, 42004135, 42004132) and State Key Laboratory of Lunar and Planetary Science (SKL-LPS(MUST)-2021-2023).

**Data Availability Statement:** The $F_{10.7}$ index and solar wind and IMF data are provided by the OMNI database (https://omniweb.gsfc.nasa.gov/ (accessed on 1 March 2022)), and the AE, Kp, and Dst indices are provide by the World Data Center, Kyoto (http://wdc.kugi.kyoto-u.ac.jp/ (accessed on 1 March 2022)). The SABER are accessible from the SABER website: http://saber.gats-inc.com/data.php (accessed on 1 March 2022). Simulation data, simulation codes, and analysis routines are being preserved on the NCAR High-Performance Storage System (https://www2.cisl.ucar.edu/resources/storage-and-file-systems/hpss (accessed on 1 March 2022)).

**Acknowledgments:** We thank TIMED/SABER science teams for providing the data used in this study. We also would like to thank high-performance computing support from Cheyenne provided by NCAR's Computational and Information Systems Laboratory. The National Center for Atmospheric Research is sponsored by the National Science Foundation. We also thank Delores Knipp for her assistance in revising this paper.

**Conflicts of Interest:** The authors declare no conflict of interest.

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
