# Peer review of "Significant Variations of Thermospheric Nitric Oxide Cooling during the Minor Geomagnetic Storm on 6 May 2015"

_universe, doi:10.3390/universe8040236_

Round 1

Reviewer 1 Report

Reviewer’s comments to the paper “Significant Variations of Thermospheric…”

by Zheng et al.

            The paper studies the changes in the NO cooling of the thermosphere during a minor geomagnetic storm of May 6, 2015 (Dst(min) = -28 nT; the storm is rated as a minor geomagnetic storm on the NOAA storm scale). The NO cooling rate was estimated form the SABER measurements and was compared to the results of simulation by the TIEGCM model.

            In the Introduction, the authors present a brief review of the problem of the thermospheric cooling during and after magnetic storms and give a proper credit to the previous studies of the problem.

            In Subsection 3.1, the authors provide a complete enough description of the solar and geospace parameters during the storm of May 6, 2015.   

            The main parameter measured by the SABER satellite was the NO infrared emission at 5.3 μm that is considered as a main indicator of the thermospheric NO cooling. It was recalculated into the NO cooling rate in W/m3.

            The comparison of the measured and simulated cooling rates for the period of May 5-7, 2015 is presented in Fig. 2. It illustrates the first main result of the study: the peak NO cooling rate in the daytime (~1330 LT) derived from SABER measurements is substantially higher (5.3 10-8 W/m3) than the TIEGCM simulations (2.5 × 10-8 W/m3) for the considered period. In other words, TIEGCM greatly underestimates the peak NO cooling rate during the particular weak geomagnetic storm.

            Figure 3 shows the altitude of the peak NO cooling rate during May 5-7, 2015 observed by SABER and simulated by TIEGCM. The figure is aimed to illustrate the second main result of the study: the observations demonstrate a strong decrease (up to 40 km) in the height of the peak NO cooling rate from high latitudes to low latitudes, whereas there is no such decrease in the simulation data. In other words, the model “…failed to capture the altitude variations of peak NO cooling rate during this weak geomagnetic storm.” There is found also a significant difference in the altitude of the maximum NO cooling rate between the SABER measurements and TIEGCM simulations at local noon at low and mid latitudes during quiet geomagnetic conditions.

            In a more visual way, the aforementioned difference for a latitude of 20 S is illustrated in Fig. 4. Figure 5 shows the vertical profiles of the NO cooling rate obtained from the SABER data and TIEGCM outputs from ~77°S to ~20°S for the quiet conditions and storm-time period. Figure 5 illustrates the third main conclusion of the study: the character of the change in the magnitude and vertical profile of the NO cooling rate from quiet conditions to the storm-time conditions is different if the SABER data and TIEGCM simulations are compared.   

            The description of the above results is directly followed by the sentence: “These results indicate that the storm-induced horizontal transport of NO density from high latitudes to lower latitudes became relatively weak at low latitudes and moved downward to lower altitude and thus caused the peak altitude decrease of NO cooling rate”. I think that this conclusion “hangs in the air”. It is worth specifying the relation of the results in the previous paragraph to that strong statement.   

            To support the conclusion on the changes in the storm-induced horizontal transport, the authors present Fig. 6. It shows the neutral density inferred from SWARM observations at 450 km for the period of 5-7 May 2015. The authors state that the observed significant density enhancements during the minor geomagnetic storm “… indicate that the equatorward winds induced by the minor geomagnetic storm were unexpectedly strong that not only can reach the low latitudes but also last until the day after the storm’. In my mind, the conclusion is not properly grounded. There could be other mechanism of thermospheric density changes during geomagnetic disturbances (for example, a mechanism related to internal waves of various scales). It is not also evident that the increase in the atmospheric density and NO content should be the same. I advise either to reject the part with the SWARM data, or to provide a stronger grounding to the aforementioned conclusion on the role of horizontal winds.  

            Overall, the paper contains important results on the behavior of the NO cooling in the thermosphere during a weak geomagnetic storm. I think that the three main results of the study mentioned in this review are quite enough for considering the paper pertinent for publication. However, I recommend a minor revision according to my comments.

            A small comment: in the capture to Figure 4, the latitude should be either 20 S, or – 20o.

Reviewer 2 Report

In this paper the authors present satellite measurements of the 5.3 micrometer radiation from NO over 3 days in which a minor geomagnetic storm occurs near the start of the second day. They present a plot of emission over a range of latitudes as a function of time. This shows some high latitude response after the onset of the storm, then enhanced emission over high to low latitudes about 20 hours later. Individual profiles of emission versus altitude are shown for selected times at a low latitude and for selected latitudes at a quiet time and another during the storm. The authors interpret the results as showing unexpectedly high equatorward winds, produced by the storm, carrying NO to low latitudes. 

As "unexpectedly high winds" is a main conclusion, the authors should share with the reader the magnitudes they are referring to. I would like to know what is an expected wind speed that they are comparing with, what is the wind speed that they estimate from the data, and is there evidence (in the literature) that such high winds occur in the ionosphere?

It is not clear to me from reading the paper how the authors deduce the presence of equatorward wind from their data. If wind was moving NO from latitude -60 to -20, I would expect to see a trail in figure 3 trending towards the right and upwards with time. Instead the main NO enhancement seems to occur simultaneously at all latitudes. The only feature in Figure 3 that looks like evidence for wind is the trace starting at about -60 degrees at day 6.3 and ending at -80 degrees at day 6.5, i.e. a poleward wind.

While the authors claim that Figure 6 shows the same evidence for equatorwind winds, I again can see no trailing to the right in the figure that would constitute such evidence. However, at 450 km altitude, I wonder if molecular diffusion could explain the apparently very rapid transfer of gas to the low latitudes. The authors should consider if molecular diffusion could be significant.

Minor points:

While the English expression is tolerable, the use of wrong words such as "responds patterns" and the frequent incorrect omission or inclusion of connecting words such as "the" means that the reader has to work out from context what is meant. However, the only point I cannot work out is the phrase "thus merged the normal peak".

I presume that "1330 LT" means "13:30 LT"?

Why is the sentence "Li et al. (2018, 2019) investigated
the model thermospheric temperature and winds responding to intense geomagnetic storms at mid and low latitudes." in the Conclusions? This is not a conclusion, nor does it add support to a conclusion.
